# LEARNING GENERALIZED HAMILTONIAN DYNAMICS WITH STABILITY FROM NOISY TRAJECTORY DATA

## ABSTRACT

We propose a unified framework for learning generalized Hamiltonian dynamics from noisy, sparse phase-space observations via variational Bayesian inference. Modeling conservative, dissipative, and port-Hamiltonian regimes with a single architecture is challenging because each induces distinct phase-space behavior from similar initial energies. To address this, we extend sparse symplectic Gaussian processes with random Fourier features to build a probabilistic surrogate of the Hamiltonian landscape. Our approach supports all three regimes through a generalized formulation of Hamiltonian dynamics. To ensure physical correctness and stability, we softly enforce conservation and Lyapunov-style constraints. With this relaxation, we recast the original constrained optimization problem as a hyperparameter-balanced multi-term loss trained end-to-end. Experiments on standard Hamiltonian benchmarks show improved long-horizon forecasting, stronger short-term prediction accuracy, and higher physics conformity compared with state-of-the-art methods.

## 1 INTRODUCTION

Learning dynamical systems from data is a fundamental problem in physics-informed machine learning, with widespread applications in physics, engineering, and robotics (Heinonen et al., 2018; Chen et al., 2018; Ayed et al., 2019; Manek & Kolter, 2019). The aim in these works and our work is to learn operators which map from initial condition to trajectories. This has been considered for deterministic systems, and also stochastic systems (Liu et al., 2019; Kidger et al., 2021). We focus on various forms of generalized Hamiltonian dynamics, conservative (Greydanus et al., 2019; Rath et al., 2021; Ross & Heinonen, 2023), dissipative (Sosanya & Greydanus, 2022; Zhong et al., 2020; Tanaka et al., 2022), and port-Hamiltonian (van der Schaft & Jeltsema, 2014; Desai et al., 2021). The different classes of dynamics present different phase spaces which must be learned. While there can be complex phase spaces even for conservative systems, adding dissipation and/or external forcing introduces additional complexity, such as the overlapping of trajectories. We present slightly different methodologies for the different classes of dynamics, but with a commonality of learning the phase space by stochastic process-based training with enforcement of stability.

Simple sparse variational regression techniques in Euclidean space often fail to accurately capture the dynamics with low uncertainty in prediction as a result of the lack of utilizing the Hamiltonian manifold and missing the underlying physical constitutive and conservation laws in space. The method of choice is to learn dynamics through optimization on Hamiltonian manifolds, (in this case Riemannian) and in a principled way that achieves stability throughout the dynamics learning process. In this work, we utilize the differential properties of the Hamiltonian manifold, its symplectic form, while preserving the conservation laws and stability properties as implicit soft regularizers, allowing us to learn dynamics in a manner consistent with inherent physical principles, thereby going beyond what standard Euclidean regression can achieve.

We focus on differentially learning stochastic Hamiltonian systems, which describe the stochastic evolution of states in phase space, comprising generalized positions $\mathbf{q}(t) \in \mathbb{R}^d$ and generalized conjugate momenta $\mathbf{p}(t) \in \mathbb{R}^d$ while exploiting both the symplectic energy conservation form and the dissipative forms, caused by motion in dissipative environments. These dynamics are governed by a generalized Hamiltonian function $\mathcal{H}(\mathbf{x}(t))$, where $\mathbf{x}(t) = [\mathbf{q}(t); \mathbf{p}(t)] \in \mathbb{R}^{2d}$ represents the state of

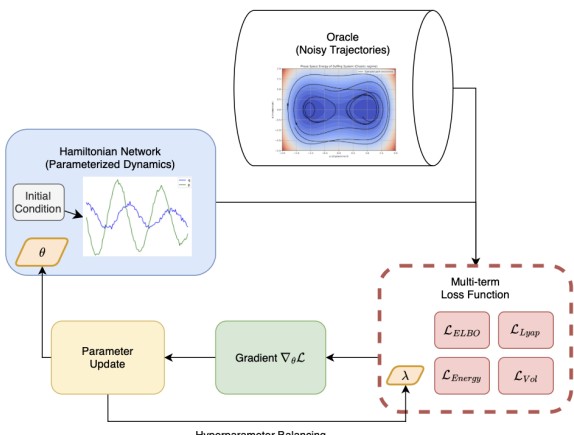

Figure 1: A visualization of our framework for generalized Hamiltonian learning. The Hamiltonian network queries data, which is used to evaluate the quality of the model through the multi-term loss function. The gradients of the loss are update the Hamiltonian dynamics parameters and the process is repeated.

the system. The principled learning of stochastic Hamiltonians are estimated by path-wise spectrally kernelized (random Fourier feature) stochastic Gaussian processes. We show the advantages of differential dynamical modeling with generalized Hamiltonian systems and Hamiltonian gradients.

Our contributions include:

- We develop a probabilistic framework for learning Hamiltonian dynamics that is robust to real-world scenarios with noisy and incomplete data.
- We extend the optimization problem by incorporating Lyapunov stability and conservation law constraints, enabling the learned dynamics to adhere closely to physical principles. By relaxing constraints and balancing multiple losses, we turn the original intractable problem into a computationally feasible training method.
- Our experiments on standard and generalized Hamiltonian systems demonstrate the improved performance both in terms of accuracy and physical validity compared to prior state-of-the-art methods.

## 2 RELATED WORKS

**Generalized Hamiltonian Dynamics**    Hamiltonian dynamics describe the evolution of position $\mathbf{q}$ and conjugate momentum $\mathbf{p}$ via a Hamiltonian function $\mathcal{H}(\mathbf{q}, \mathbf{p})$ that defines the system's energy landscape. In standard Hamiltonian dynamics, generalized coordinates $(\mathbf{q}, \mathbf{p})$ evolve along constant-energy trajectories determined by $\mathcal{H}$. To account for many physical phenomena, several generalizations incorporate dissipation and external forcing terms (van der Schaft & Jeltsema, 2014):

- Conservative Hamiltonian dynamics: $(\dot{\mathbf{q}}, \dot{\mathbf{p}}) = J\nabla\mathcal{H}(\mathbf{q}, \mathbf{p})$.
- Dissipative Hamiltonian dynamics: $(\dot{\mathbf{q}}, \dot{\mathbf{p}}) = (J + D)\nabla\mathcal{H}(\mathbf{q}, \mathbf{p})$.
- Port-Hamiltonian dynamics: $(\dot{\mathbf{q}}, \dot{\mathbf{p}}) = (J + D)\nabla\mathcal{H}(\mathbf{q}, \mathbf{p}) + F(t)$.

### 2.1 GENERALIZED HAMILTONIAN DYNAMICS LEARNING

Several works have attempted to learn or recover underlying Hamiltonian dynamics from datasets of trajectories of the generalized coordinates $(\mathbf{q}, \mathbf{p})$ and their time derivatives $(\dot{\mathbf{q}}, \dot{\mathbf{p}})$. In particular, the Hamiltonian Neural Network (HNN) (Greydanus et al., 2019) learns conservative Hamiltonian dynamics by parameterizing $\mathcal{H}$ with a deterministic neural network and minimizing an MSE loss on

time-derivative data. While HNN applies to conservative systems, extensions such as dissipative HNN (D-HNN) (Sosanya & Greydanus, 2022) target dissipative systems by learning a second network to represent the dissipative component of the dynamics. Port-HNN (P-HNN) (Desai et al., 2021) parameterizes neural networks for the Hamiltonian, the dissipation matrix, and the external forcing function to learn dynamics within the Port-Hamiltonian framework (van der Schaft & Jeltsema, 2014), which is the most general of these settings. Nonetheless, these methods often require not only $(\mathbf{q}, \mathbf{p})$ trajectories but also $(\dot{\mathbf{q}}, \dot{\mathbf{p}})$ trajectories. More importantly, the parameterized networks are deterministic and therefore struggle with probabilistic data

## 2.2 Probabilistic Learning of Generalized Hamiltonian Dynamics

Gaussian Processes (GPs) have been widely used for modeling dynamical systems with uncertainty, particularly under sparse or noisy observations (Heinonen et al., 2018). GPs have been applied to learning general ODE dynamics (Bhouri & Perdikaris, 2022) and to Hamiltonian systems (Tanaka et al., 2022; Ross & Heinonen, 2023). To address computational limitations, Random Fourier Features (RFF) (Rahimi & Recht, 2007) provide efficient GP approximations, with further refinements such as Sparse Spectrum GPs (Lázaro-Gredilla et al., 2010) and Spherical Structured Features (Lyu, 2017). However, generic GP approximations often fail to capture the crucial symplectic structure of Hamiltonian systems. Methods such as Symplectic Gaussian Process Regression (SymGPR) (Rath et al., 2021), Symplectic Spectrum Gaussian Processes (SSGP) (Tanaka et al., 2022), and Hamiltonian Gaussian Processes (HGP) (Ross & Heinonen, 2023) address this by incorporating symplecticity into GP-based models of noisy dynamics. SymGPR uses GP regression directly, while SSGP and HGP are trained to maximize the evidence lower bound (ELBO) on trajectory data. Gaussian Process Port-Hamiltonian Systems (GPPHS) (Beckers et al., 2022) extend this approach to learning Port-Hamiltonian dynamics with dissipation and external forcing.

However, enforcing symplecticity alone is insufficient. In this work, we explicitly incorporate physical laws, including stability and conservation principles, into learning across different classes of Hamiltonian dynamics. While Hamiltonian systems admit symmetries and invariants such as energy and phase-space volume (Meiss, 2007), these hold strictly in the conservative case; for dissipative dynamics, energy and volume decay over time, and for Port-Hamiltonian systems, their evolution depends on external power injection. Accordingly, we adopt different, physics-informed training treatments for each case, ensuring that the learned dynamics respect the appropriate physical constraints in conservative, dissipative, and Port-Hamiltonian settings.

## 3 Methods

### 3.1 Problem Statement

We are given a dynamics dataset $\mathcal{D}$ with $I$ trajectories of phase–space observations. The $i$-th trajectory provides data $\{(\mathbf{y}_{ij}, t_{ij})\}_{j=1}^{J_i}$, where $\mathbf{y}_{ij} \in \mathbb{R}^{2d}$ are noisy measurements of the latent phase–space coordinates $\mathbf{x}_{ij} = (\mathbf{q}_{ij}, \mathbf{p}_{ij})$ at time $t_{ij}$. Our aim is to learn the underlying dynamics and in particular the Hamiltonian $\mathcal{H}(\mathbf{q}, \mathbf{p})$. Let $\theta$ denote all parameters of the Hamiltonian dynamics model. The naive unconstrained estimator would maximize the marginal log–likelihood $\log p(\mathcal{D} \mid \theta)$.

However, beyond likelihood fit, we consider a more challenging problem of requiring the learned vector field to respect core physical invariants including energy conservation, phase-space volume, and dynamical stability. In particular, for conservative Hamiltonian systems, two canonical invariants are energy conservation and phase–space volume conservation (Liouville's theorem) (Meiss, 2007).

**Energy conservation.** The total energy is constant along trajectories:
$$\mathcal{H}\big(\mathbf{q}(t), \mathbf{p}(t)\big) = \mathcal{H}\big(\mathbf{q}(0), \mathbf{p}(0)\big). \tag{1}$$

**Volume conservation (Liouville).** For any measurable set $A$ in phase space and flow map $\rho_t$,
$$\int_{\rho_t(A)} \mathbf{1} \, d\mathbf{q} \, d\mathbf{p} = \int_A \mathbf{1} \, d\mathbf{q} \, d\mathbf{p}. \tag{2}$$

**Lyapunov stability.** For an ODE $x' = f(x)$, Lyapunov stability asserts the existence of $V$ with $V(x^*) = 0$, $V(x) > 0$ for $x \neq x^*$, and $\frac{d}{dt} V(x(t)) \leq 0$; asymptotic stability strengthens this to

$< 0$, while $\alpha$–exponential stability requires $\frac{d}{dt}V(x(t)) \leq -\alpha V(x(t))$ together with $c_1\|x-x^*\| \leq V(x) \leq c_2\|x-x^*\|$. In Hamiltonian settings, we may take $V = \mathcal{H}$, and set $\alpha = 0$.

**Physics–aware objective.** Incorporating these physical laws yields a constrained optimization problem:

$$
\max_\theta p(\mathcal{D} \mid \theta)
$$

$$
\begin{aligned}
\text{s.t. } &(\dot{\mathbf{q}}_i, \dot{\mathbf{p}}_i) = (J + D)\nabla\mathcal{H}(\mathbf{q}_i, \mathbf{p}_i) + F(t),\\
&\big(\mathcal{H} \circ \rho_t\big)(\mathbf{q}_{i0}, \mathbf{p}_{i0}) = \mathcal{H}(\mathbf{q}_{i0}, \mathbf{p}_{i0}),\\
&\int_{\rho_t(A)} \mathbf{1}d\mathbf{q}d\mathbf{p} = \int_A \mathbf{1}d\mathbf{q}d\mathbf{p},\\
&\tfrac{d}{dt}\mathcal{H}(\mathbf{q}_{ij}, \mathbf{p}_{ij}) \leq -\alpha\,\mathcal{H}(\mathbf{q}_{ij}, \mathbf{p}_{ij}),\\
&\mathcal{H}(\mathbf{q}_{ij}, \mathbf{p}_{ij}) \geq 0,\\
&\forall\, i = 1{:}I,\ j = 0{:}J_i.
\end{aligned}
\tag{3}
$$

Problem (3) is generally intractable due to latent trajectories, nonlinear ODE constraints, and hard physical constraints. In this section, we introduce a novel physics-informed relaxation probabilistic method that makes the problem both computationally solvable and robust to noisy data. We proceed in two steps:

1. **Probabilistic Hamiltonian Surrogate Model:** Model $\mathcal{H}$ as a stochastic process and replace the intractable marginal likelihood with a tractable evidence lower bound (ELBO).
2. **Physics-constrained learning:** Derive relaxation versions for hard constraints in (3) cast the unconstrained optimization problem through Lagrangian multipliers into regularized training loss function.

### 3.2 Probabilistic Hamiltonian Surrogate Model

To learn a conservative, dissipative, and Port-Hamiltonian dynamics in a systematic scheme, we propose to decouple energy conservation term $J\nabla\mathcal{H}$, dissipative term $D\nabla\mathcal{H}$, and external forces $F(t)$, when applicable. Our scheme treats each module individually and enforces additional constraints to ensure that each term captures desired physical property from noisy observations. We start by stating our choice of stochastic Hamiltonian surrogate, which is a random Fourier feature (RFF) approximation of a Gaussian process (GP).

Our stochastic Hamiltonian follows a GP prior:
$$
\begin{aligned}
\mathcal{H}(\mathbf{q}, \mathbf{p}) &\sim GP(0, K((\mathbf{q}, \mathbf{p}), (\mathbf{q}', \mathbf{p}'))),\\
\nabla\mathcal{H}(\mathbf{q}, \mathbf{p}) &\sim GP(0, \tilde{K}), \quad \tilde{K} = \nabla^2 K.
\end{aligned}
\tag{4}
$$

The specific GP prior we choose is the spectral kernel with random Fourier features (RFF) (Tanaka et al., 2022).

$$
\mathcal{H}(\mathbf{q}, \mathbf{p}) = \sum_{m=1}^{M} w_m^\top \phi_m(\mathbf{q}, \mathbf{p}), \quad \phi_m(\mathbf{x}) = \begin{bmatrix} \cos(\omega_m^\top \mathbf{x}) \\ \sin(\omega_m^\top \mathbf{x}) \end{bmatrix}.
\tag{5}
$$

The weights $w_m$ are approximated by normal distribution with prior $p(w_m) = \mathcal{N}(0, \sigma_0^2\mathbf{I}_2)$ and posterior $p(w_m|\mathcal{D}) = N(b, C)$ with parameters $\sigma_0^2$, $b$ and $\sqrt{C}$, where $\sqrt{C}$ is used instead to enforce positive semi-definiteness of $C$. The frequencies are sampled $\omega_m \sim \mathcal{N}(0, \Lambda^{-1})$ where $\Lambda$ is a diagonal matrix as an additional parameter. The RFF basis representation approximates a GP prior with Gaussian kernel function. To obtain the Hamiltonian gradient $\nabla H$, one can differentiate the RFF basis analytically:

$$
\begin{aligned}
\nabla\mathcal{H}(\mathbf{q}, \mathbf{p}) &= \sum_{m=1}^{M} w_m^\top \nabla\phi_m(\mathbf{q}, \mathbf{p}),\\
\nabla\phi_m(\mathbf{x}) &= \begin{bmatrix} -\sin(\omega_m^\top \mathbf{x})\omega_m^\top \\ \cos(\omega_m^\top \mathbf{x})\omega_m^\top \end{bmatrix}.
\end{aligned}
\tag{6}
$$

PARAMETERIZATIONS FOR DIFFERENT DYNAMICS

Based on this RFF representation of Hamiltonian energy function, we demonstrate how to parameterize three different classes of Hamiltonian dynamics:

**Conservative Hamiltonian Systems**    For the conservative case, the set of learnable parameters $\theta$ is composed of those for the GP-RFF surrogate and for data uncertainty. The surrogate is defined by the RFF prior variance $\sigma_0^2$, posterior parameters $b$ and $\sqrt{C}$, and length-scale matrix $\Lambda$. The uncertainty is modeled by the standard deviation of the initial condition, $a$, and the observation noise, $\sigma$. The complete parameter set is therefore: $\theta = (\sigma_0, b, \sqrt{C}, \Lambda, a, \sigma)$.

**Dissipative Hamiltonian Systems**    For the case of dissipative Hamiltonian dynamics, additional parameters are needed for the dissipation term, specifically the matrix $D$. The dissipation matrix is not directly parameterized since it has a certain structure. We assume the structure of a diagonal matrix with the first half of the diagonal (affecting $\mathbf{q}$) being zero and the second half of the diagonal (affecting $\mathbf{p}$) being non-positive. Thus, the form is $D = diag(0, \cdots, 0, D_1, \cdots, D_d), D_i \leq 0$. To enforce non-positivity of $D_i$, we use a parameter $\eta \in \mathbb{R}^d$ and set $D_i = -\eta_i^2$. Thus, the full parameter array for learning dissipative Hamiltonian dynamics is: $\theta = (\sigma_0, b, \sqrt{C}, \Lambda, a, \sigma, \eta)$.

**Port-Hamiltonian Systems**    For port Hamiltonian dynamics the external forcing term $F(t)$ must be parameterized. We assume forcing directly effects the momentum only and thus $F(t) = [0 \ F_\vartheta(t)]^T$, where the parameterized forcing function (for $p$) is $F_\vartheta(t)$ with parameters $\vartheta$. We set $F_\vartheta$ to be an MLP neural network. The full parameter array for learning port-Hamiltonian dynamics is: $\theta = (\sigma_0, b, \sqrt{C}, \Lambda, a, \sigma, \eta, \vartheta)$.

GENERATIVE MODEL AND VARIATIONAL INFERENCE

For each trajectory $i$ we draw an initial phase space state $\mathbf{x}_0^{(i)} \sim p(\mathbf{x}_0) = \mathcal{N}(\mathbf{0}, a^2\mathbf{I})$, integrate the parameterized vector field to obtain dynamical state $\mathbf{x}_{ij} = \Phi_\theta(t_{ij}; \mathbf{x}_0^{(i)})$, and obtain noisy observations $\mathbf{y}_{ij} \sim \mathcal{N}(\mathbf{x}_{ij}, \sigma^2\mathbf{I})$, $j = 1, \ldots, J_i$. Let $\mathbf{W}$ denote the collection of RFF weights. With the usual conditional independence assumptions the joint distribution factorizes as

$$p(\mathcal{D}, \mathbf{X}, \mathbf{W}) = \Big(\prod_{i,j} p(\mathbf{y}_{ij} \mid \mathbf{x}_{ij})\Big)\Big(\prod_i p(\mathbf{x}_0^{(i)})\Big)p(\mathbf{W}),$$

where $\mathbf{X}$ stacks all phase space states along all trajectories. We choose the mean–field variational family $q(\mathbf{W}, \mathbf{X}_0) = q(\mathbf{W}) \prod_i q(\mathbf{x}_0^{(i)})$ with $q(\mathbf{W}) = \prod_m \mathcal{N}(\mathbf{w}_m \mid \mathbf{b}_m, \mathbf{C}_m)$ and $q(\mathbf{x}_0^{(i)}) = \mathcal{N}(\boldsymbol{\mu}_i, \mathrm{diag}\,\boldsymbol{\sigma}_i^2)$. Standard manipulations of maximization of joint distribution log likelihood yield the evidence lower bound (ELBO) loss

$$\mathcal{L}_{\mathrm{ELBO}} = \sum_{i=1}^{I} \sum_{j=1}^{J_i} \mathbb{E}_q\big[\log \mathcal{N}(\mathbf{y}_{ij} \mid \mathbf{x}_{ij}, \sigma^2\mathbf{I})\big] \tag{7}$$
$$- \mathrm{KL}\big(q(\mathbf{W})\|p(\mathbf{W})\big) - \sum_{i=1}^{I} \mathrm{KL}\big(q(\mathbf{x}_0^{(i)})\|p(\mathbf{x}_0)\big).$$

In practice, the expectation in the ELBO is estimated via Monte Carlo: we sample $\mathbf{w} \sim q(\mathbf{w})$ and $\boldsymbol{\omega} \sim q(\boldsymbol{\omega})$, draw each initial state $\mathbf{x}_{i0} \sim q(\mathbf{x}_{i0})$, propagate it deterministically through $\Phi_\theta$ using a differentiable ODE solver (Chen et al., 2018), and average the resulting log-likelihoods of $\{\mathbf{y}_{ij}\}$.

## 3.3 PHYSICS-CONSTRAINED LEARNING

Having defined the surrogate model with the ELBO function $\mathcal{L}_{\mathrm{ELBO}}$ replacing the usual log-likelihood, we now proceed to introduce the relaxation constraints to make (3) computationally solvable.

**Energy conservation:** The energy conservation law can be softly enforced with the following additional loss term:
$$\mathcal{L}_{\mathrm{Energy}} = \mathbb{E}_{\mathbf{q}, \mathbf{p}, t}\left[\mathcal{H}(\rho_t(\mathbf{q}, \mathbf{p})) - \mathcal{H}(\mathbf{q}, \mathbf{p})\right]^2. \tag{8}$$

This constraint term can be estimated by evaluating on the trajectories which were used to compute the ELBO loss in the previous section.

**Volume conservation:** The law of conservation of volume can be discretized and evaluated through the following loss term:

$$\mathcal{L}_{\text{Vol}} = \frac{1}{N} \left[ \sum_{n=1}^{N} \Big( \mathbf{1}_A(\mathbf{q}_n, \mathbf{p}_n) - \mathbf{1}_A(\rho_t(\mathbf{q}_n, \mathbf{p}_n)) \Big) \right]^2, \tag{9}$$

where $\mathbf{1}_A$ refers the characteristic function of a sampled "volume" $A$. In implementation we sample a rectangular domain $A$ in each epoch and the loss is computed under different sampled time stamps.

**Lyapunov stability:** For this physical constraint, we use the following Lyapunov loss function based on prior works (Manek & Kolter, 2019; Rodriguez et al., 2022) as a relaxation term:

$$\mathcal{L}_{\text{Lyap}} = \int_0^\top \lambda_{1,1} \text{ReLU} \left( \frac{d}{dt} H(x(t)) \right) dt + \int_0^T \lambda_{1,2} \text{ReLU}(-H(x(t))) dt. \tag{10}$$

### 3.4 Optimization and Training

By replacing maximum–likelihood with ELBO minimization and relaxing hard physics constraints, we convert the original intractable program (3) into a tractable *soft* objective via a Lagrangian formulation with multipliers $\lambda$:

$$\mathcal{L}(\theta, \boldsymbol{\lambda}) = \mathcal{L}_{\text{ELBO}}(\theta) + \lambda_1 \mathcal{L}_{\text{Lyap}}(\theta) + \lambda_2 \mathcal{L}_{\text{Energy}}(\theta) + \lambda_3 \mathcal{L}_{\text{Vol}}(\theta). \tag{11}$$

The vector $\boldsymbol{\lambda}$ is adapted during training using the gradient descent / ascent (GDA) scheme described in the next paragraph. When all $\lambda_k = 0$, the method reduces to pure variational inference. When $\lambda_k$ are large, the optimization approaches a hard-constrained formulation.

#### Balancing the Multi-Term Loss

The loss consists of various terms, which must be carefully balanced in order to prevent under- or over-constrained optimization: Hyperparameters $\boldsymbol{\lambda}$ can be balanced by solving a max-min optimization problem for both the parameters and hyperparameters:

$$\min_\theta \max_{\boldsymbol{\lambda}} \mathcal{L}(\theta, \boldsymbol{\lambda}). \tag{12}$$

Both can be updated using gradient-based optimization methods (SGD, Adam, etc.), but for the hyperparameters we use ascent rather than descent since it is a maximization problem. This is a variant of the Gradient Descent-Ascent (GDA) (Lin et al., 2020; Zhang et al., 2020):

$$\theta_{k+1} = \theta_k - g_1 \left( \frac{\partial \mathcal{L}}{\partial \theta} \right), \qquad \boldsymbol{\lambda}_{k+1} = \boldsymbol{\lambda}_k + g_2 \left( \frac{\partial \mathcal{L}}{\partial \boldsymbol{\lambda}} \right), \tag{13}$$

where $g_1, g_2$ depend on the optimizer used. Note that the gradient of the loss with respect to the hyperparameters can be directly computed since the loss is linear with respect to the hyperparameters. In our experiments, we compare the use of no balancing (all lambdas equal to one) with GDA-based balancing. We test two variants of GDA. The first uses Adam (Kingma & Ba, 2015) for the parameter update and gradient ascent for the hyperparameters. The second uses Adam for both updates.

**Alternative optimization methods.** We consider MTAdam (Malkiel & Wolf, 2020) as an alternative way of balancing multiple loss terms. Rather than solving a min-max problem with updates to $\lambda$, MTAdam groups the parameters and within each group normalizes the gradients of the individual loss terms. Jacobian descent (Quinton & Rey, 2024) is another method, which computes the Jacobian of the loss vector with respect to the parameter vector, and then applies one of several aggregators to the Jacobian. Aggregators such as UPGrad are meant to construct the update such that each individual loss is improved, rather than one loss being improved at the expense of another. See the appendix for additional results on various balancing methods.

TRAINING ALGORITHM

The complete training loop is summarized in Algorithm 1. We learn the model parameters by obtaining batches of noisy trajectory data and updating the parameters based on the gradient of the total balanced loss function.

---

**Algorithm 1: Learning Generalized Hamiltonian Dynamics**

---

**Require:** Dataset $\mathcal{D}$, epochs $K$, batch size $B$, horizon $T$, learning rate $\alpha$
1: Initialize parameters $\theta_0$          ▷ Sections 3.2
2: **for** $k = 1$ to $K$ **do**
3:      Sample $\{\mathbf{y}_t^{(b)} = (\mathbf{q}_t^{(b)}, \mathbf{p}_t^{(b)})\}_{b=1,t=1}^{B,T}$ from $\mathcal{D}$
4:      **for** $b = 1$ to $B$ **do**
5:          Initialize $\mathbf{x}_0^{(b)} = (\mathbf{q}_0^{(b)}, \mathbf{p}_0^{(b)})$.
6:          Sample RFF weights $w \sim \mathcal{N}(b, \sqrt{C})$.
7:          Sample RFF frequencies $\omega \sim \mathcal{N}(0, \Lambda^{-1})$.
8:          Construct Hamiltonian $\mathcal{H}$, its gradient $\nabla\mathcal{H}$, and vector field $f$.    ▷ Sections 3.2
9:          Integration step: $\{\mathbf{x}_t^{(b)}\}_{t=1}^T \leftarrow \mathrm{odeint}(\mathbf{x}_0^{(b)}, f, T)$.
10:     **end for**
11:     Compute total loss $\mathcal{L}(\theta_k, \boldsymbol{\lambda}_k)$.          ▷ Section 3.4
12:     Compute loss gradient $\nabla_\theta \mathcal{L}(\theta_k, \boldsymbol{\lambda}_k)$.
13:     Parameter update: $\theta_{k+1} \leftarrow \mathrm{Adam}(\theta_k, \nabla_\theta \mathcal{L}, \alpha)$
14:     Hyperparameter update: $\boldsymbol{\lambda}_{k+1} \leftarrow \boldsymbol{\lambda}_k + \alpha\nabla_{\boldsymbol{\lambda}}\mathcal{L}$    ▷ Section 3.4
15: **end for**
16: **return** Optimized parameters $\theta_K$

---

| Class | System | (D,P)HNN | SSGP | Ours (Equal) | Ours (GDA) |
|-------|--------|----------|------|--------------|------------|
| Cons. | Single Pendulum | $0.5922 \pm 1.4887$ | $1.0186 \pm 0.8825$ | $\mathbf{0.1404 \pm 0.2166}$ | $0.2834 \pm 0.3872$ |
| Diss. | Damped Spring | $0.0347 \pm 0.0640$ | $0.0275 \pm 0.0135$ | $\mathbf{0.0269 \pm 0.0127}$ | $0.0273 \pm 0.0130$ |
| Diss. | Damped Pendulum | $0.3312 \pm 0.4097$ | $0.1377 \pm 0.1917$ | $\mathbf{0.0902 \pm 0.1270}$ | $0.0929 \pm 0.1021$ |
| Diss. | Unforced Duffing | $0.1790 \pm 0.5278$ | $0.0904 \pm 0.3435$ | $0.0879 \pm 0.2996$ | $\mathbf{0.0876 \pm 0.2914}$ |
| Port | Forced Spring | $0.0923 \pm 0.1362$ | $\mathbf{0.0293 \pm 0.0102}$ | $0.0311 \pm 0.0126$ | $0.0319 \pm 0.0129$ |
| Port | Forced Duffing | $0.4127 \pm 0.7770$ | $0.4920 \pm 1.0299$ | $\mathbf{0.2742 \pm 0.2930}$ | $0.2763 \pm 0.2602$ |

Table 1: Evaluation on testing datasets of 6 systems of 3 Hamiltonian classes. We compare HNN variants, SSGP, and our methods. We use HNN (Greydanus et al., 2019), DHNN (Sosanya & Greydanus, 2022), or PHNN (Desai et al., 2021) depending on the associated class of system. For our method, we compare equally weighted loss terms to GDA-balanced terms. We compare using the metric of the MSE loss of the $q, p$ testing trajectories. In each system besides forced spring, the MSE loss is lower for our method, with the equal weighting typically performing better.

| Class | System | (D,P)HNN | SSGP | Ours (Equal) | Ours (GDA) |
|-------|--------|----------|------|--------------|------------|
| Cons. | Conservative Spring | $0.3335 \pm 0.3539$ | $0.1930 \pm 0.3596$ | $\mathbf{0.1142 \pm 0.2208}$ | $0.1359 \pm 0.2508$ |
| Cons. | Single Pendulum | $2.2017 \pm 2.8207$ | $\mathbf{1.5811 \pm 1.9176}$ | $1.5851 \pm 1.8575$ | $1.5844 \pm 1.8625$ |
| Diss. | Damped Pendulum | $0.5563 \pm 0.6734$ | $0.1309 \pm 0.1885$ | $\mathbf{0.1046 \pm 0.1599}$ | $0.1803 \pm 0.2445$ |
| Diss. | Damped Spring | $0.1335 \pm 0.1191$ | $0.0047 \pm 0.0028$ | $\mathbf{0.0033 \pm 0.0019}$ | $0.0039 \pm 0.0021$ |
| Diss. | Unforced Duffing | $0.0346 \pm 0.2420$ | $0.0296 \pm 0.2765$ | $\mathbf{0.0088 \pm 0.0057}$ | $0.0092 \pm 0.0058$ |
| Port | Forced Duffing | $91.8247 \pm 321.6358$ | $1.5024 \pm 1.6520$ | $0.9603 \pm 0.6543$ | $\mathbf{0.8232 \pm 0.5024}$ |
| Port | Forced Spring | $6.0665 \pm 0.3321$ | $6.2233 \pm 0.2051$ | $6.0434 \pm 0.0781$ | $\mathbf{6.0348 \pm 0.0803}$ |
| Port | Windy Pendulum | $1.1722 \pm 1.2881$ | $0.4936 \pm 0.2704$ | $0.3712 \pm 0.1890$ | $\mathbf{0.3208 \pm 0.1789}$ |

Table 2: Evaluation of long-term forecasting on various methods. The metric is MSE $\pm$ standard deviation of $(q, p)$ trajectories. For long-term forecasting, the trajectories are 5 times the horizon of the trajectories in the training datasets ($[0, 50]$ instead of $[0, 10]$).

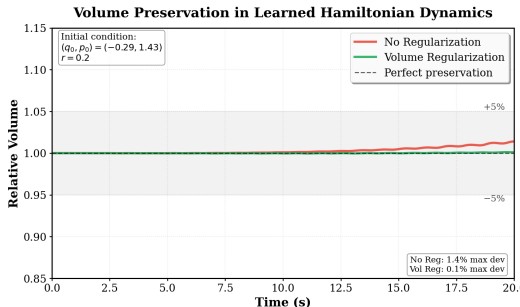

Figure 2: Volume preservation comparison between unregularized and volume-regularized models. Starting from initial condition $(q_0, p_0) = (2.0, -2.0)$ with a circular region of radius $r = 0.2$, we integrate the boundary forward in time using the learned dynamics with dissipation set to zero ($\eta = 0$). The unregularized model (red) exhibits significant volume contraction, while the volume-regularized model (green) maintains better preservation of phase space volume, demonstrating the effectiveness of the volume preservation regularizer $\mathcal{L}_{\text{vol}}$ in learning divergence-free dynamics. The dashed line indicates perfect volume preservation, and the shaded region represents $\pm 5\%$ deviation.

## 4 EXPERIMENTS

### 4.1 EXPERIMENT SETUPS

**Tasks and Datasets.** We use standard tasks for Hamiltonian dynamics learning (Greydanus et al., 2019; Sosanya & Greydanus, 2022; Desai et al., 2021; Tanaka et al., 2022). These tasks cover all 3 classes of Hamiltonian systems: standard conservative systems, dissipative systems, and port-Hamiltonian system. The dataset for each task consists of 100 training noisy trajectories and 100 testing trajectories, where each trajectory contains 100 datapoints with timestamps within $[0, 10]$. The specific systems are summarized in Section A, with dataset configurations in Table 4.

**Baselines.** We evaluate our methods against state-of-the-art approaches: HNN (Greydanus et al., 2019), DHNN (Sosanya & Greydanus, 2022), P-HNN (Desai et al., 2021), and SSGP (Tanaka et al., 2022). We also consider different variants of our approach using regularized loss with conservation and stability corrections. Additionally, we compare different versions of our method of balancing individual loss terms. In particular, we consider an equally weighted loss function versus using GDA for updating $\lambda$ weights.

**Hyperparameter choices.** The batch size is set to 100, and the learning rate is $10^{-3}$. For all models including baselines and our models' variants, we use 5000 epochs. Each epoch uses the full dataset. The SSGP (Tanaka et al., 2022) model uses 100 RFF basis functions. The HNN models use one hidden layer with 200 hidden nodes, with the HNN (Greydanus et al., 2019) having 1 network $\mathcal{H}$, the DHNN (Sosanya & Greydanus, 2022) having 2 networks $\mathcal{H}, \mathcal{D}$, and the PHNN (Desai et al., 2021) having 2 networks $\mathcal{H}, F$ and a dissipation matrix $D$. For port-Hamiltonian systems, the SSGP model is supplemented by an external forcing function $F(t)$, which is a neural network with one hidden layer of 100 hidden nodes. The configurations of the datasets are given in Table 4. The configurations for the unforced and forced Duffing systems are based on Desai et al. (2021).

**Compute.** Experiments were performed on NVIDIA Grace CPU with 116 GB RAM and NVIDIA H200 GPU with 96 GB VRAM, using Linux operating system and PyTorch (Paszke et al., 2019) machine learning library. We use torchdiffeq (Chen et al., 2018) library for the numerical integration function odeint in Algorithm 1.

### 4.2 RESULTS

The results are listed in Table 1, as well as visualized in Figure 4. In Figure 5 we show additional comparisons of learned phase space maps for port-Hamiltonian systems. From Table 1, and Figure 4, it can be seen that our method can achieve improved performance across systems in each of the various classes of generalized Hamiltonian dynamics.

**Long-Range Forecasting** To demonstrate the benefit of conservative and stability regularization on generalization, we evaluate our method against prior methods on long-range forecasts. Here the models are evaluated on trajectories in the time domain $[0, 50]$, while they were trained on $[0, 10]$. The results can be seen in Table 2, and show improved generalization to long time horizons for our methods.

**Validation of Conservative Property** To demonstrate the effect of the conservation constraint regularizers, we display a volume calculation along time steps in Figure 2. The unregularized SSGP model (red curve) does not preserve volume and $\nabla \cdot J\nabla\mathcal{H}$ term deviates starting from unseen integration times ($t \geq 10$), a clear indication that the learned vector field acquires a spurious non zero divergence. This behavior demonstrates that the regularizer effectively enforces a divergence-free flow, preventing artificial volume loss and yielding a faithful symplectic approximation even without explicit dissipative terms.

## 4.3 ABLATION STUDIES

Ablation study experiment results are available in the supplamentary material. In Table 8, we compare the use of different loss terms for single, damped, and windy pendulum systems. We compare the unregularized ELBO against applying each regularizer individually and combined. We compare both equal and GDA-balanced weighting of loss terms. In Tables 5, 6, 7 and Figure 6, we compare on datasets of various noise levels, ranging from noiseless to highly noisy data. We compare for each of the three classes of Hamiltonian dynamics.

## 5 CONCLUSION

We present a unified framework for learning generalized Hamiltonian dynamics from noisy data by combining a probabilistic Hamiltonian surrogate with physics-aware constraints and balanced optimization. For conservative, dissipative, and port-Hamiltonian systems, the method produces accurate trajectories with strong physical fidelity. In the future, we plan to extend this framework to a control setting in order to support uncertainty decision making and make the framework applicable to a wider range of domains.

### REPRODUCIBILITY STATEMENT

Additional details of the methodology are in the supplementary material in Section C. Details of the configurations of the datasets are in Section A. The code and data is available at `https://anonymous.4open.science/r/HamiltonianLearning-8CCA` and in the supplementary material.

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

USE OF LARGE LANGUAGE MODELS (LLMS)

All content of the paper was written by the authors. LLMs were used for the aid of code implementation, formatting LaTeX tables/figures, and spelling/grammar checking.

## A  HAMILTONIAN SYSTEMS

We list out 9 Hamiltonian systems in Table 3. They are grouped based on the three classes described in the previous section. The phase space dynamics of these systems are visualized in Figure 3. The parameters we use in the generation of the datasets for these systems are given in Table 4.

| System Name | Hamiltonian | Euler-Lagrangian Equation | (q,p)-Dynamics (type) |
|:---:|:---:|:---:|:---:|
| Single Pendulum | P | $l\ddot{q} + g\sin q = 0$ | $J\nabla\mathcal{H}$ |
| Simple Spring | S | $m\ddot{q} + kq = 0$ | $J\nabla\mathcal{H}$ |
| Henon-Heiles | HH | $\begin{cases} \ddot{q}_1 + q_1 + 2q_1q_2 = 0 \\ \ddot{q}_2 + q_2 - (q_1^2 - q_2^2) = 0 \end{cases}$ | $J\nabla\mathcal{H}$ |
| Damped Pendulum | DP | $l\ddot{q} + \gamma\dot{q} + g\sin q = 0$ | $(J + D)\nabla\mathcal{H}$ |
| Damped Spring | DS | $m\ddot{q} + \gamma\dot{q} + kq = 0$ | $(J + D)\nabla\mathcal{H}$ |
| Unforced Duffing Equation | UD | $\ddot{q} + \gamma\dot{q} + \alpha q + \beta q^3 = 0$ | $(J + D)\nabla\mathcal{H}$ |
| Windy Pendulum | WP | $l\ddot{q} + \gamma\dot{q} + g\sin q = F(t)$ | $(J + D)\nabla\mathcal{H} + \bar{F}(t)$ |
| Forced Spring | FS | $m\ddot{q} + \gamma\dot{q} + kq = F(t)$ | $(J + D)\nabla\mathcal{H} + F(t)$ |
| Duffing Equation | DE | $\ddot{q} + \gamma\dot{q} + \alpha q + \beta q^3 = F(t)$ | $(J + D)\nabla\mathcal{H} + F(t)$ |

Table 3: Different (generalized) Hamiltonian systems with their corresponding Euler-Lagrangian equations are grouped into conservative, dissipative, and port-Hamiltonian classes, each having a form of $(q, p)$ dynamics.

The Hamiltonian energy functions are given below. Note that for the different types of pendulum, spring, and Duffing systems, they share same physical model, and therefore their Hamiltonians are identical, but the dynamics are different depending on dissipation/forcing.

$$\mathcal{H}(q,p) = \frac{p^2}{2ml^2} - mgl\cos q \qquad \text{(P/DP/WP)}$$

$$\mathcal{H}(q,p) = \frac{1}{2}kq^2 + \frac{p^2}{2m} \qquad \text{(S/DS/FS)}$$

$$\mathcal{H}(q_1,q_2,p_1,p_2) = \frac{1}{2}(p_1^2 + p_2^2) + \frac{1}{2}(q_1^2 + q_2^2) + q_1^2q_2 - \frac{1}{3}q_2^3 \qquad \text{(HH)}$$

$$\mathcal{H}(q,p) = \frac{p^2}{2m} + \frac{\alpha q^2}{2} + \frac{\beta q^4}{4} \qquad \text{(UD/DE)}$$

The Lagrangian of each problem is:

$$\mathcal{L}(q,\dot{q}) = \frac{1}{2}ml^2\dot{q}^2 + mgl\cos q \qquad \text{(P/DP/WP)}$$

$$\mathcal{L}(q,\dot{q}) = \frac{1}{2}m\dot{q}^2 - \frac{1}{2}kq^2 \qquad \text{(S/DS/FS)}$$

$$\mathcal{L}(q_1,q_2,\dot{q}_1,\dot{q}_2) = \frac{1}{2}(\dot{q}_1^2 + \dot{q}_2^2) - \frac{1}{2}(q_1^2 + q_2^2) - q_1^2q_2 + \frac{1}{3}q_2^3 \qquad \text{(HH)}$$

$$\mathcal{L}(q,\dot{q}) = \frac{1}{2}m\dot{q}^2 - \frac{\alpha q^2}{2} - \frac{\beta q^4}{4} \qquad \text{(UD/DE)}$$

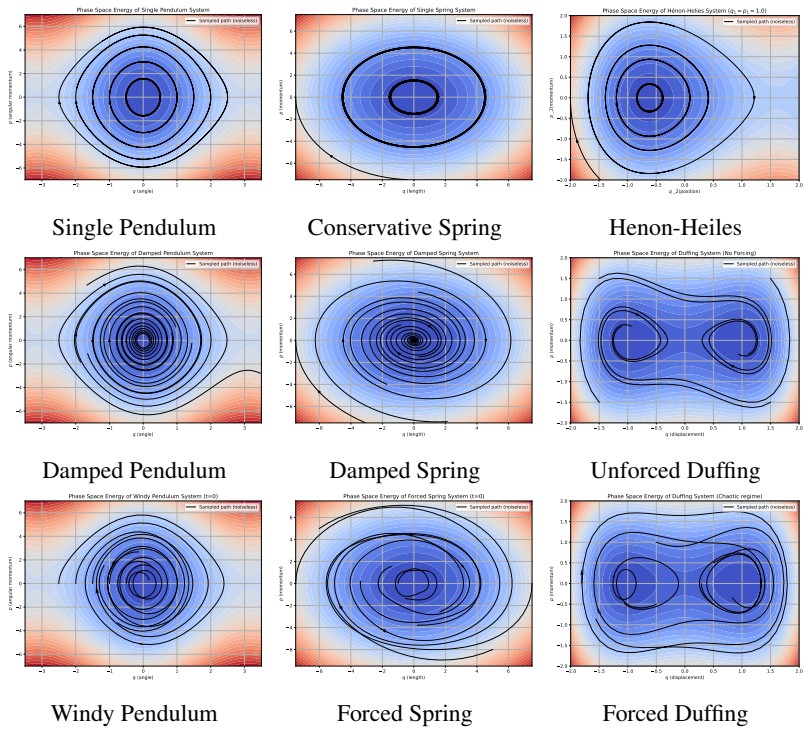

| Single Pendulum | Conservative Spring | Henon-Heiles |
| --- | --- | --- |
| Damped Pendulum | Damped Spring | Unforced Duffing |
| Windy Pendulum | Forced Spring | Forced Duffing |

Figure 3: Phase map diagrams with sampled trajectories for various conservative (top row), dissipative (middle row), and port-Hamiltonian (bottom row) systems. In these examples, the heat map denotes the Hamiltonian energy, and the dynamics of systems are plotted for various initial conditions. Column 1 shows pendulum systems, column 2 shows spring systems, and column 3 shows the Henon-Heiles system for the conservative case and Duffing systems for the dissipative and port cases.

The corresponding phase space dynamics are given accordingly, with time-dependent forces being defined:

$$\begin{bmatrix} \dot{q} \\ \dot{p} \end{bmatrix} = \begin{bmatrix} 0 & 1 \\ -1 & 0 \end{bmatrix} \begin{bmatrix} mgl \sin q \\ p/ml^2 \end{bmatrix} \tag{dqdp-P}$$

$$\begin{bmatrix} \dot{q} \\ \dot{p} \end{bmatrix} = \begin{bmatrix} 0 & 1 \\ -1 & -\gamma \end{bmatrix} \begin{bmatrix} mgl \sin q \\ p/ml^2 \end{bmatrix} \tag{dqdp-DP}$$

$$\begin{bmatrix} \dot{q} \\ \dot{p} \end{bmatrix} = \begin{bmatrix} 0 & 1 \\ -1 & -\gamma \end{bmatrix} \begin{bmatrix} mgl \sin q \\ p/ml^2 \end{bmatrix} + \begin{bmatrix} 0 \\ vt \end{bmatrix} \tag{dqdp-WP}$$

$$\begin{bmatrix} \dot{q} \\ \dot{p} \end{bmatrix} = \begin{bmatrix} 0 & 1 \\ -1 & 0 \end{bmatrix} \begin{bmatrix} kq \\ p/m \end{bmatrix} \tag{dqdp-S}$$

$$\begin{bmatrix} \dot{q} \\ \dot{p} \end{bmatrix} = \begin{bmatrix} 0 & 1 \\ -1 & -\gamma \end{bmatrix} \begin{bmatrix} kq \\ p/m \end{bmatrix} \tag{dqdp-DS}$$

$$\begin{bmatrix} \dot{q} \\ \dot{p} \end{bmatrix} = \begin{bmatrix} 0 & 1 \\ -1 & -\gamma \end{bmatrix} \begin{bmatrix} kq \\ p/m \end{bmatrix} + \begin{bmatrix} 0 \\ F_0 \sin(\omega t) \sin(2\omega t) \end{bmatrix} \tag{dqdp-FS}$$

$$\begin{bmatrix} \dot{q_1} \\ \dot{q_2} \\ \dot{p_1} \\ \dot{p_2} \end{bmatrix} = \begin{bmatrix} 0 & I \\ -I & 0 \end{bmatrix} \begin{bmatrix} q_1 + 2q_1 q_2 \\ q_2 + q_1^2 - q_2^2 \\ p_1 \\ p_2 \end{bmatrix} \tag{dqdp-HH}$$

$$\begin{bmatrix} \dot{q} \\ \dot{p} \end{bmatrix} = \begin{bmatrix} 0 & 1 \\ -1 & -\gamma \end{bmatrix} \begin{bmatrix} \alpha q + \beta q^3 \\ p/m \end{bmatrix} \tag{dqdp-UD}$$

$$\begin{bmatrix} \dot{q} \\ \dot{p} \end{bmatrix} = \begin{bmatrix} 0 & 1 \\ -1 & -\gamma \end{bmatrix} \begin{bmatrix} \alpha q + \beta q^3 \\ p/m \end{bmatrix} + \begin{bmatrix} 0 \\ F_{ext} \end{bmatrix} \tag{dqdp-DE}$$

| Dataset | Type | Trajs. | Steps | Noise | Damping | Forcing | Other Params |
|---|---|---|---|---|---|---|---|
| Single Pendulum | Cons. | 100 | 100 | 0.1 | N/A | N/A | $m=1, g=9.81, \ell=1$ |
| Damped Pendulum | Diss. | 100 | 100 | 0.1 | 0.1 | N/A | $m=1, g=9.81, \ell=1$ |
| Windy Pendulum | Port | 100 | 100 | 0.01 | 0.1 | 0.1t | $m=1, g=9.81, \ell=1$ |
| Mass-Spring | Cons. | 100 | 100 | 0.1 | N/A | N/A | $m=1, k=1$ |
| Damped Spring | Diss. | 100 | 100 | 0.1 | 0.1 | N/A | $m=1, k=1$ |
| Forced Spring | Port | 100 | 100 | 0.1 | 0.1 | $0.1\sin(t)\sin(2t)$ | $m=1, k=1$ |
| Henon-Heiles | Cons. | 100 | 100 | 0.1 | N/A | N/A | $m=1, \alpha=1$ |
| Unforced Duffing | Diss. | 100 | 100 | 0.1 | 0.3 | N/A | $\alpha=-1, \beta=1$ |
| Forced Duffing | Port | 100 | 100 | 0.1 | 0.1 | 0.39 | $\alpha=-1, \beta=1$ |

Table 4: Parameters for the setup of the various Hamiltonian dynamics trajectory datasets.

# B  ADDITIONAL RESULTS

In Figures 4 and 5 we compare the errors in the phase maps of the learned Hamiltonian energy contours. In Tables 5, 6, 7 and Figure 6, we evaluate our methods with various loss term balancing methods against prior works. These evaluations are done on various noise levels ranging from noiseless to Gaussian noise with 0.2 standard deviation. While methods such as HNN (Greydanus et al., 2019) may perform better for cases with little to no noise, our methods perform better in high noise scenarios. Note also that HNN,and varients require derivative information, that is $\dot{q}, \dot{p}$ data, while SSGP (Tanaka et al., 2022) and our methods do not. This is especially relevant when data is noisy and/or sparse, since finite difference approximations will be much less reliable in those cases. In Table 8, we show an ablation study of our method using the addition of only one regularizer or all regularizers, for both equal and GDA loss term balancing.

| Method | Noise level $\sigma$ | | | | |
|---|---|---|---|---|---|
| | **0** | **0.01** | **0.05** | **0.1** | **0.2** |
| *Prior works* | | | | | |
| SSGP | **0.0028±0.0017** | 0.0039±0.0019 | 0.0108±0.0063 | **0.0242±0.0126** | 0.0933±0.0524 |
| HNN | *0.0031±0.0028* | **0.0032±0.0027** | 0.0131±0.0579 | 0.0678±0.1300 | 0.3253±0.1300 |
| *Ours* | | | | | |
| Equal | 0.0041±0.0015 | 0.0044±0.0023 | *0.0083±0.0046* | 0.0307±0.0161 | 0.0918±0.0591 |
| GDA | 0.0044±0.0016 | 0.0046±0.0024 | 0.0087±0.0048 | 0.0311±0.0164 | 0.0893±0.0534 |
| MTAdam | 0.2116±0.2044 | 0.2011±0.1669 | 0.2880±0.3215 | 0.2470±0.2517 | 0.2736±0.2337 |
| JD | 0.0041±0.0015 | 0.0044±0.0023 | *0.0083±0.0047* | 0.0307±0.0161 | 0.0917±0.0590 |
| JD2 | 0.0036±0.0019 | *0.0035±0.0020* | **0.0077±0.0052** | *0.0271±0.0160* | *0.0889±0.0489* |
| Noise Prior | 0.0148±0.0110 | 0.0072±0.0054 | 0.0191±0.0126 | 0.0292±0.0183 | **0.0859±0.0495** |

Table 5: Comparison of various noise levels evaluated on the testing dataset for the conservative spring system. We test with uncorrupted data and datasets with Gaussian noise of standard deviation 0.01, 0.05, 0.1, and 0.2. We compare the base SSGP model (Tanaka et al., 2022) to our method with various ways of weighting the multi-term loss function, which are equal summation, gradient descent-ascent (GDA), multi-term Adam (MTAdam) (Malkiel & Wolf, 2020), Jacobian descent with a UPGrad aggregator (Quinton & Rey, 2024) with the combined ELBO (JD) and separated ELBO (JD2), and GDA with prior knowledge of noise. The metric is the MSE loss on the $q, p$ testing trajectories. Here our method performs the best in the majority of cases, but the type of loss balancing that is optimal varies depending on the noise level.

# C  ADDITIONAL METHODOLOGY DETAILS

## C.1  NUMERICAL RECIPE OF REGULARIZED LOSS TERM

**Conservation of Energy**  To compute the ELBO loss, we already generate trajectories for different initial conditions. We can evaluate to Hamiltonian along these trajectories and check the difference

| Method | Noise level $\sigma$ | | | | |
|---|---|---|---|---|---|
| | **0** | **0.01** | **0.05** | **0.1** | **0.2** |
| *Prior works* | | | | | |
| SSGP | 0.0477±0.0717 | 0.0405±0.0531 | 0.0664±0.0815 | 0.0897±0.1240 | 0.1521±0.1844 |
| DHNN | **0.0048±0.0052** | **0.0047±0.0057** | *0.0406±0.0578* | 0.3844±0.5541 | 0.8012±0.8763 |
| *Ours* | | | | | |
| Equal | 0.0365±0.0536 | 0.0470±0.0606 | 0.0927±0.1102 | 0.1097±0.1511 | **0.1284±0.1267** |
| GDA | *0.0316±0.0447* | 0.0363±0.0491 | 0.0456±0.0636 | 0.1119±0.1540 | 0.1376±0.1354 |
| MTAdam | 0.4440±0.4672 | 0.4377±0.4635 | 0.4440±0.4672 | 0.4584±0.4642 | 0.5071±0.4716 |
| JD | 0.0377±0.0510 | 0.0458±0.0593 | 0.0957±0.1140 | *0.0734±0.1015* | *0.1286±0.1268* |
| Noise Prior | 0.0876±0.1157 | *0.0273±0.0332* | **0.0312±0.0465** | **0.0559±0.0693** | 0.1633±0.2005 |

Table 6: Comparison of various noise levels evaluated on the testing dataset for the damped pendulum system. We test with uncorrupted data and datasets with Gaussian noise of standard deviation $0.01, 0.05, 0.1$, and $0.2$. We compare the base SSGP model (Tanaka et al., 2022) to our method with various ways of weighting the multi-term loss function, which are equal summation, gradient descent-ascent (GDA), multi-term Adam (MTAdam) (Malkiel & Wolf, 2020), Jacobian descent (JD) with a UPGrad aggregator (Quinton & Rey, 2024), and GDA with prior knowledge of noise. The metric is the MSE loss on the $q, p$ testing trajectories. Here, by applying additional regularization with proper balancing to enforce physical constraints and stability, we achieve superior performance in each case of noise level. In addition, in most cases having a prior knowledge of the noise level further improves the performance.

| Method | Noise level $\sigma$ | | | | |
|---|---|---|---|---|---|
| | **0** | **0.01** | **0.05** | **0.1** | **0.2** |
| *Prior works* | | | | | |
| SSGP | 0.2419±0.2798 | 0.2377±0.2776 | 0.2600±0.3145 | 0.2530±0.2869 | 0.2924±0.2902 |
| PHNN | **0.0145±0.0278** | **0.0247±0.0636** | *0.1643±0.2686* | 0.3484±0.5941 | 0.7203±3.1528 |
| *Ours* | | | | | |
| Equal | 0.2438±0.2732 | 0.2869±0.4264 | 0.2816±0.3729 | 0.2559±0.2743 | *0.2739±0.1703* |
| GDA | 0.2471±0.2814 | *0.2470±0.2908* | *0.2565±0.2940* | *0.2344±0.2256* | **0.2738±0.1708** |
| MTAdam | 0.2532±0.1351 | 0.2419±0.1236 | 0.2524±0.1215 | 0.2502±0.1280 | 0.2921±0.1371 |
| JD | 0.2629±0.3136 | 0.2972±0.4204 | 0.3127±0.4624 | 0.2561±0.2898 | 0.2863±0.2640 |
| JD2 | 0.2900±0.4033 | 0.2865±0.4033 | 0.2577±0.3058 | 0.2445±0.2661 | 0.2951±0.3025 |
| Noise Prior | 0.2847±0.3471 | 0.3241±0.6112 | 0.2603±0.3035 | 0.2958±0.5060 | 0.2976±0.2760 |
| Noise Prior 2 | *0.0628±0.1531* | *0.0474±0.1132* | **0.1054±0.1824** | **0.1857±0.2483** | 0.2906±0.3187 |

Table 7: Comparison of various noise levels evaluated on the testing dataset for the chaotic duffing system. We test with uncorrupted data and datasets with Gaussian noise of standard deviation $0.01, 0.05, 0.1$, and $0.2$. We compare the base SSGP model (Tanaka et al., 2022) to our method with various ways of weighting the multi-term loss function, which are equal summation, gradient descent-ascent (GDA), multi-term Adam (MTAdam) (Malkiel & Wolf, 2020), Jacobian descent with a UPGrad aggregator (Quinton & Rey, 2024) with the combined ELBO (JD) and separated ELBO (JD2), and GDA with prior knowledge of noise (Noise Prior) and its modification learning only the mean path (Noise Prior 2). The metric is the MSE loss on the $q, p$ testing trajectories. Here the method using our multi-term loss function with the a prior noise level, typically performs the best, most likely because it accurately separates noise from the true dynamics while accounting for physical constraints.

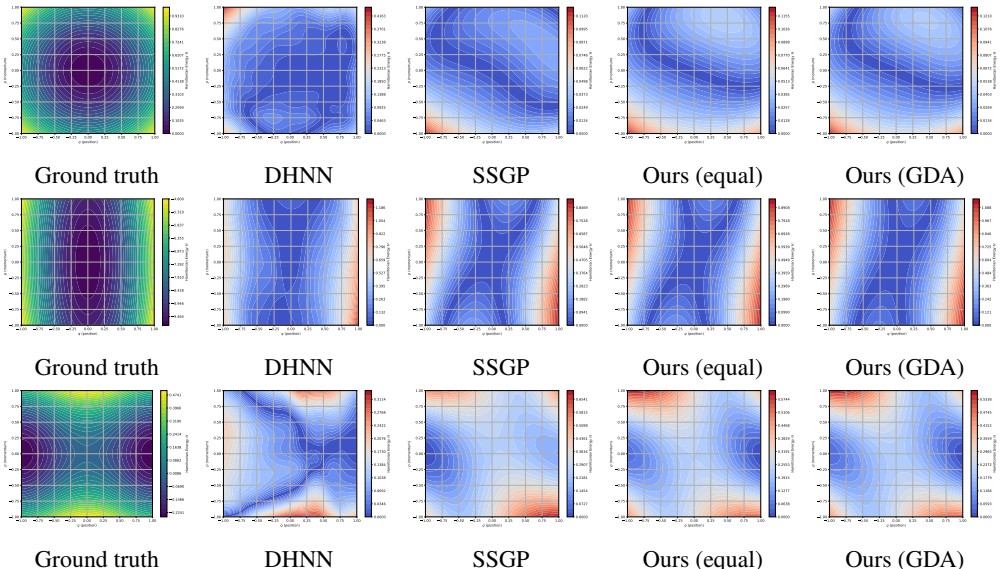

Figure 4: The 1st column is the true phase maps with Hamiltonian energy contours three different dissipative Hamiltonian systems: damped spring (top row), damped pendulum (middle row), and unforced Duffing (bottom row). Columns 2 through 5 shows the error between the true Hamiltonian energy and learned Hamiltonian energy for both prior work and our methods. The prior work is shown in column 2 with DHNN (Sosanya & Greydanus, 2022) and column 3 with SSGP (Tanaka et al., 2022). Our method is shown in column 4, using equally weighted loss terms, and column 5, using GDA-balanced loss terms.

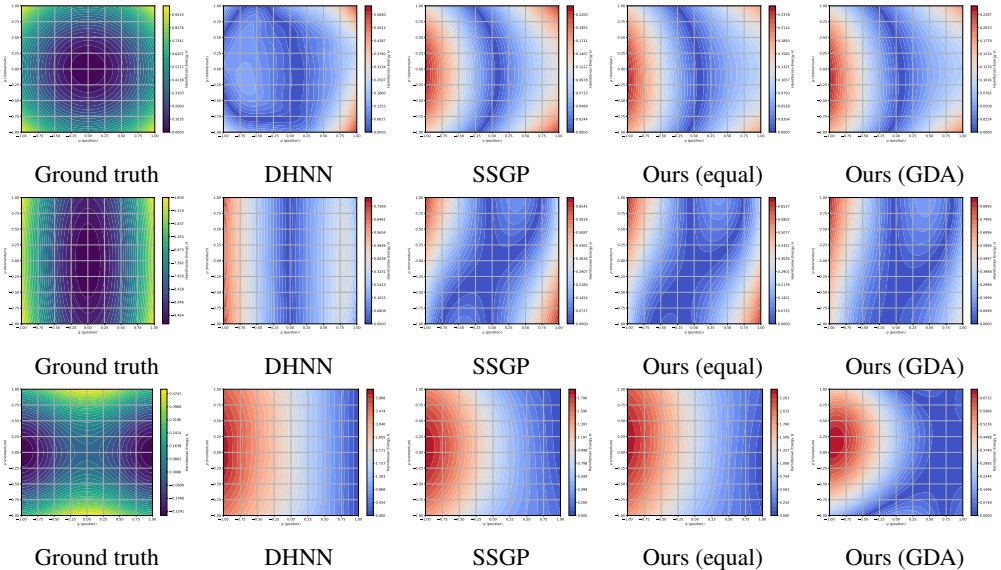

Figure 5: The 1st column is the true phase maps with Hamiltonian energy contours three different port-Hamiltonian systems: forced spring (top row), windy pendulum (middle row), and forced Duffing (bottom row). Columns 2 through 5 shows the error between the true Hamiltonian energy and learned Hamiltonian energy for both prior work and our methods. The prior work is shown in column 2 with PHNN (Desai et al., 2021) and column 3 with SSGP (Tanaka et al., 2022). Our method is shown in column 4, using equally weighted loss terms, and column 5, using GDA-balanced loss terms.

| Dataset | equal | equal_E | equal_L | equal_V | gda | gda_E | gda_L | gda_V | noreg |
|---|---|---|---|---|---|---|---|---|---|
| Single Pendulum | 0.1111 | **0.0941** | **0.0941** | 0.1315 | 0.1634 | 0.5146 | 0.2812 | 0.1315 | 1.0222 |
| Damped Pendulum | 0.0714 | 0.1622 | 0.1629 | 0.0671 | **0.0613** | 0.1004 | 0.1347 | 0.0671 | 0.1733 |
| Windy Pendulum | 0.0670 | 0.3019 | 0.1113 | 0.4420 | 0.2972 | **0.0303** | 0.0829 | 0.4420 | 0.0493 |

Table 8: Mean squared error (MSE) over test sets for single (top row), damped (middle row), and windy (bottom row) pendulum systems. We compare the use of each loss term individually and the combination of all terms, using either equal and GDA-balanced weighting, against unregularized ELBO. Bold indicates the lowest MSE in each row. E denotes only the energy regularizer is used, while V and L denote the volume and Lyapunov regularizers respectively.

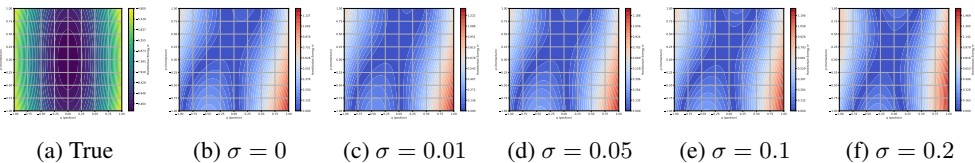

| (a) True | (b) $\sigma = 0$ | (c) $\sigma = 0.01$ | (d) $\sigma = 0.05$ | (e) $\sigma = 0.1$ | (f) $\sigma = 0.2$ |

Figure 6: Phase map comparison for the damped pendulum system with various noise levels. Subfigure (a) shows the ground truth Hamiltonian. Subfigure (b) show the error in learned Hamiltonian from an uncorrupted dataset. Subfigures (c-f) show the error in the learned Hamiltonian from datasets corrupted by Gaussian noise with the specific standard deviation. All of these Hamiltonians were learned using our GDA-balanced method.

from initial time. Thus, the term may be estimated as follows:

$$\mathcal{L}_{Energy} \approx \sum_{i=1}^{I} \sum_{j=1}^{J} \sum_{k=1}^{N_x} \frac{1}{I J_i N_x} \left( \mathcal{H}^\theta \left( \mathbf{x}_{ij}^{(k)} \right) - \mathcal{H}^\theta \left( \mathbf{x}_{i0}^{(k)} \right) \right)^2. \tag{14}$$

Here $N_x$ refers number of GP process realization of $\mathcal{H}^\theta$. Note that these conservation law constraint terms can be estimated by evaluating on the trajectories which were generated to compute the ELBO loss in the previous section.

**Conservation of Volume** The loss term is

$$\mathcal{L}_{\text{Vol}} = \frac{1}{N} \left[ \sum_{n=1}^{N} \left( \mathbf{1}_A(\mathbf{q}_n, \mathbf{p}_n) - \mathbf{1}_A(\rho_t(\mathbf{q}_n, \mathbf{p}_n)) \right) \right]^2, \tag{15}$$

where $\mathbf{1}_A$ refers the characteristic function of a sampled "volume" $A$. In implementation we sample a rectangular domain $A$ in each epoch and the loss is computed under different sampled time stamps.

**Lyapunov Stability** In order to estimate this, the loss can be computed along the same trajectories as were generated for the computation of the ELBO loss and also used for the conservation of energy and volume. This estimation is the following:

$$\mathcal{L}_{Lyap} \approx \sum_{i=1}^{I} \sum_{j=1}^{J} \sum_{k=1}^{N_x} \left[ \lambda_{1,1} ReLU \left( \frac{\mathcal{H}^\theta \left( \mathbf{x}_{i,j+1}^{(k)} \right) - \mathcal{H}^\theta \left( \mathbf{x}_{i,j}^{(k)} \right)}{\Delta t} \right) + \lambda_{1,2} ReLU \left( -\mathcal{H}^\theta \left( \mathbf{x}_{i,j}^{(k)} \right) \right) \right] \tag{16}$$

In practice, since the first term is already satisfied by enforcing conservation of energy we omit it in the code implementation.

DEPENDENCE OF LOSS TERMS ON PARAMETER GROUPS

We have several groups of parameters as well as a multi-term loss function. The parameters required in the computation of each loss function are as follows:

- ELBO: $\sigma_0, b, \sqrt{C}, \Lambda, a, \sigma, \eta, \vartheta$ (all)

- – KL-divergence (initial condition): $a$
- – KL-divergence (RFF weights): $b, \sqrt{C}, \sigma_0$
- – Negative log-liklihood: $b, \sqrt{C}, \sigma, \Lambda, \eta, \vartheta$
- Conservation of Energy: $b, \sqrt{C}, \Lambda$
- Conservation of Volume: $b, \sqrt{C}, \Lambda$
- Lyapunov Stability: $b, \sqrt{C}, \Lambda, \eta, \vartheta$

From this we can construct the block-matrix Jacobian.

$$
J = \begin{bmatrix}
\mathbf{0} & \mathbf{0} & \mathbf{0} & \mathbf{0} & \dfrac{\partial \mathcal{L}_{\text{KL0}}}{\partial a} & \mathbf{0} & \mathbf{0} & \mathbf{0} \\[2mm]
\dfrac{\partial \mathcal{L}_{\text{KLW}}}{\partial \sigma_0} & \dfrac{\partial \mathcal{L}_{\text{KLW}}}{\partial b} & \dfrac{\partial \mathcal{L}_{\text{KLW}}}{\partial \sqrt{C}} & \mathbf{0} & \mathbf{0} & \mathbf{0} & \mathbf{0} & \mathbf{0} \\[2mm]
\mathbf{0} & \dfrac{\partial \mathcal{L}_{\text{NLL}}}{\partial b} & \dfrac{\partial \mathcal{L}_{\text{NLL}}}{\partial \sqrt{C}} & \dfrac{\partial \mathcal{L}_{\text{NLL}}}{\partial \Lambda} & \dfrac{\partial \mathcal{L}_{\text{NLL}}}{\partial \sigma} & \mathbf{0} & \dfrac{\partial \mathcal{L}_{\text{NLL}}}{\partial \eta} & \dfrac{\partial \mathcal{L}_{\text{NLL}}}{\partial \vartheta} \\[2mm]
\mathbf{0} & \dfrac{\partial \mathcal{L}_{\text{Energy}}}{\partial b} & \dfrac{\partial \mathcal{L}_{\text{Energy}}}{\partial \sqrt{C}} & \dfrac{\partial \mathcal{L}_{\text{Energy}}}{\partial \Lambda} & \mathbf{0} & \mathbf{0} & \mathbf{0} & \mathbf{0} \\[2mm]
\mathbf{0} & \dfrac{\partial \mathcal{L}_{\text{Vol}}}{\partial b} & \dfrac{\partial \mathcal{L}_{\text{Vol}}}{\partial \sqrt{C}} & \dfrac{\partial \mathcal{L}_{\text{Vol}}}{\partial \Lambda} & \mathbf{0} & \mathbf{0} & \mathbf{0} & \mathbf{0} \\[2mm]
\mathbf{0} & \dfrac{\partial \mathcal{L}_{\text{Lyap}}}{\partial b} & \dfrac{\partial \mathcal{L}_{\text{Lyap}}}{\partial \sqrt{C}} & \dfrac{\partial \mathcal{L}_{\text{Lyap}}}{\partial \Lambda} & \mathbf{0} & \mathbf{0} & \dfrac{\partial \mathcal{L}_{\text{Lyap}}}{\partial \eta} & \dfrac{\partial \mathcal{L}_{\text{Lyap}}}{\partial \vartheta}
\end{bmatrix}
$$

## C.2 Comparison to Onsager-Machlup Functional

The Onsager-Machlup function $OM$ (Dürr & Bach, 1978; Zhang & Li, 2024) is an analog of the Lagrangian for SDEs. It has the property that a minimizer $z$ of the Onsager-Machlup functional, $\int_0^T OM(z, \dot{z}) dt$, is the most probable path for the associated SDE. The Onsager-Machlup function for an SDE of the form:

$$d\mathbf{x}_t = \mathbf{f}(\mathbf{x}) dt + d\mathbf{W}_t \tag{17}$$

is

$$OM(\mathbf{z}, \dot{\mathbf{z}}) = \frac{1}{2} \|\dot{\mathbf{z}} - \mathbf{f}(\mathbf{z})\|^2 + \frac{1}{2}(\nabla \cdot \mathbf{f}(\mathbf{z})). \tag{18}$$

For a Hamiltonian system with dissipation and noise of the form:

$$d\mathbf{x}_t = [(J + D)\nabla\mathcal{H}(\mathbf{x}) + F(t)] dt + d\mathbf{W}_t$$

the Onsager-Machlup function is:

$$OM(\mathbf{z}, \dot{\mathbf{z}}) = \frac{1}{2} \|\dot{\mathbf{z}} - [(J + D)\nabla\mathcal{H}(\mathbf{z}) + F(t)]\|^2 + \frac{1}{2}(\nabla \cdot [(J + D)\nabla\mathcal{H}(\mathbf{z}) + F(t)]) \tag{19}$$

$$= \frac{1}{2} \|\dot{\mathbf{z}} - (J + D)\nabla\mathcal{H}(\mathbf{z}) - F(t)\|^2 + \frac{1}{2}(\nabla \cdot D\nabla\mathcal{H}(\mathbf{z})) \tag{20}$$

with the antisymmetric matrix term disappearing. Thus, if there is no dissipation, then the Onsager-Machlup function reduces to an $L^2$ error. Other work has used the Onsager-Machlup functional in machine learning. In Li et al. (2021), it is used to derive a Hamiltonian system for the most probable path between the initial and terminal conditions. In Haas et al. (2014) it is used to evaluate the entropy of trajectories. Note that the first term is simply the $L^2$ loss of the predicted trajectories. The second term (the entropy term) is the divergence of the vector field. This is equivalent to conservation of volume based on Liouville's theorem, which states that volume-conserving vector fields are divergence-free (Meiss, 2007).

## C.3 Changes in Conservation Laws for Dissipative and Port-Hamiltonian Systems

The ELBO loss described for the conservative case remains the same for the dissipative and port-Hamiltonian cases, as does the Lyapunov stability loss. However, the conservation law loss must be

| Class | Physical Properties | Form of Laws | Model Parameterization ($\theta$) |
|---|---|---|---|
| Conservative | Conservation of Energy/Volume | Eq. 1, Eq. 2 | $\sigma_0, b, \sqrt{C}, \Lambda, a, \sigma$ |
| Dissipative | Dissipation of Energy/Volume | Eq. 22, Eq. 25 | $\sigma_0, b, \sqrt{C}, \Lambda, a, \sigma, \eta$ |
| Port-Hamiltonian | Dissipation/External Input of Energy/Volume | Eq. 23, Eq. 25 | $\sigma_0, b, \sqrt{C}, \Lambda, a, \sigma, \eta, \vartheta$ |

Table 9: Breakdown of the physical constraints, their numerical forms, and the model parameterization for three classes: conservative, dissipative, and port-Hamiltonian dynamics

modified since in dissipative systems energy and volume are not longer constant in time. Since the model separates the conservative, dissipative, and forcing terms of the dynamics, the conservative laws can be enforced by integrating the conservative part only, which is what our implementation does. Below we examine how energy and volume are affected by dissipative and external forcing terms.

For energy, consider the time derivative of the Hamiltonian:

$$
\begin{aligned}
\frac{d}{dt}\mathcal{H}(q(t), p(t)) &= \nabla\mathcal{H}^T(J + D)\nabla\mathcal{H} \\
&= \cancel{\nabla\mathcal{H}^T J \nabla\mathcal{H}} + \nabla\mathcal{H}^T D \nabla\mathcal{H} \\
&= -\sum_{i=1}^{d}\left(\eta_i\frac{\partial\mathcal{H}}{\partial p_i}\right)^2
\end{aligned}
\tag{21}
$$

Then the dissipation of energy can be written as

$$
\frac{d\mathcal{H}}{dt} + \sum_{i=1}^{d}\left(\eta_i\frac{\partial\mathcal{H}}{\partial p_i}\right)^2 = 0.
\tag{22}
$$

For port-Hamiltonian systems, the energy evolution can be written as

$$
\frac{d\mathcal{H}}{dt} + \sum_{i=1}^{d}\left(\eta_i\frac{\partial\mathcal{H}}{\partial p_i}\right)^2 - \nabla H^T F(t) = 0.
\tag{23}
$$

For volume, we consider the divergence of the vector field:

$$
\begin{aligned}
\nabla \cdot \mathbf{f} &= \nabla \cdot ((J + D)\nabla\mathcal{H}) \\
&= \cancel{\nabla \cdot J\nabla\mathcal{H}} + \nabla \cdot D\nabla\mathcal{H} \\
&= -\sum_{i=1}^{d}\eta_i^2\frac{\partial^2\mathcal{H}}{\partial p_i^2}.
\end{aligned}
\tag{24}
$$

Then the dissipation of volume can be written as:

$$
\nabla \cdot \mathbf{f} + \sum_{i=1}^{d}\eta_i^2\frac{\partial^2\mathcal{H}}{\partial p_i^2} = 0.
\tag{25}
$$

The port-Hamiltonian version of this is the same since the external forcing term $F(t)$ is only time-dependent and thus its divergence is zero.

### C.4 ELBO DERIVATION

Here we derive the ELBO, which follows that given in Tanaka et al. (2022). All random variables are clearly identified, and the deterministic nature of the ODE flow is handled correctly, yielding an evidence lower bound (ELBO) containing only those terms that truly require approximation.

Let $\mathcal{D} = \{\mathbf{y}_{ij}\}_{i=1:I,\, j=1:J_i}$ be the noisy observations sampled at times $\{t_{ij}\}$. The model is specified by

    1. Latent RFF weights: $\mathbf{W} \sim \mathcal{N}(\mathbf{0}, \sigma_0^2\mathbf{I})$;

    2. Latent initial states: $\mathbf{x}_{i0} \sim \mathcal{N}(\mathbf{0}, a^2\mathbf{I})$   $(i = 1, \ldots, I)$;

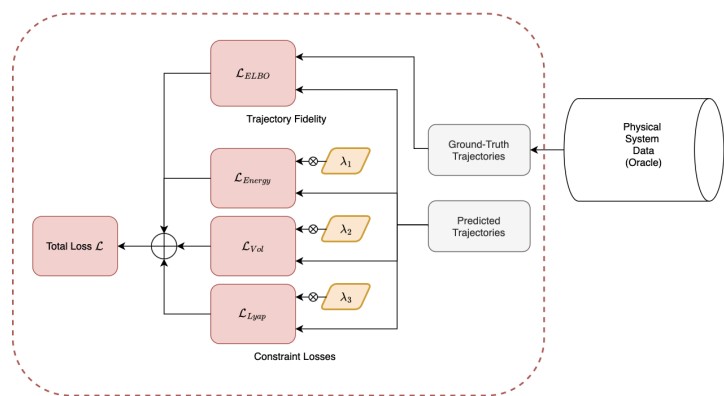

Figure 7: The breakdown of our loss function into the ELBO, energy and volume conservation, and Lyapunov stability terms. See Sections 3.4 and C.1 for details.

3. Deterministic trajectory evolution:

$$\mathbf{x}_{ij} = \Phi_{\boldsymbol{\theta}}(t_{ij}; \mathbf{x}_{i0}, \mathbf{W}),$$

where $\Phi_{\boldsymbol{\theta}}$ is the flow generated by integrating the parameterized vector field;

4. Noisy observations: $\mathbf{y}_{ij} \mid \mathbf{x}_{ij} \sim \mathcal{N}(\mathbf{x}_{ij}, \sigma^2 \mathbf{I})$.

Because every state after $t = 0$ is a deterministic function of $(\mathbf{W}, \mathbf{x}_{i0})$, the joint density factorizes as

$$p(\mathcal{D}, \mathbf{X}_0, \mathbf{W}) = \left[ \prod_{i,j} p(\mathbf{y}_{ij} \mid \mathbf{x}_{ij}) \right] \left[ \prod_i p(\mathbf{x}_{i0}) \right] p(\mathbf{W}), \tag{26}$$

with no separate probability measure required for the intermediate states $\mathbf{x}_{ij}$.

We adopt a mean-field Gaussian family for the latent variables:

$$q(\mathbf{W}, \mathbf{X}_0) = q(\mathbf{W}) \prod_{i=1}^{I} q(\mathbf{x}_{i0}), \tag{27}$$

with

$$q(\mathbf{W}) = \prod_{m=1}^{M} \mathcal{N}(\mathbf{w}_m \mid \mathbf{b}_m, \mathbf{C}_m), \qquad q(\mathbf{x}_{i0}) = \mathcal{N}(\boldsymbol{\mu}_i, \operatorname{diag} \boldsymbol{\sigma}_i^2). \tag{28}$$

No variational factor is needed for the deterministic states $\{\mathbf{x}_{ij}\}_{j \geq 1}$.

Applying Jensen's inequality to $\log p(\mathcal{D})$ and using the above factorizations gives

$$\mathcal{L}_{\text{ELBO}} = \sum_{i=1}^{I} \sum_{j=1}^{J_i} \mathbb{E}_{q(\mathbf{W})q(\mathbf{x}_{i0})} \left[ \log \mathcal{N}\big(\mathbf{y}_{ij} \mid \Phi_{\boldsymbol{\theta}}(t_{ij}; \mathbf{x}_{i0}, \mathbf{W}), \sigma^2 \mathbf{I}\big) \right]$$

$$- \operatorname{KL}\big(q(\mathbf{W}) \,\|\, \mathcal{N}(\mathbf{0}, \sigma_0^2 \mathbf{I})\big) - \sum_{i=1}^{I} \operatorname{KL}\big(q(\mathbf{x}_{i0}) \,\|\, \mathcal{N}(\mathbf{0}, a^2 \mathbf{I})\big). \tag{29}$$

Only three terms appear: a data-fit term and two KL regularizers – one for the RFF weights and one for the initial conditions. No KL term for the intermediate states occurs, as they are deterministic.

