# OpenReview forum: "Learning Generalized Hamiltonian Dynamics with Stability from Noisy Trajectory Data"
_ICLR.cc/2026/Conference — ICLR 2026 Conference Withdrawn Submission_

### Official Review · Reviewer_piGu · 2025-10-27

**Soundness:** 2
**Presentation:** 2
**Contribution:** 1
**Rating:** 2
**Confidence:** 4

**Summary:**

This paper proposes a Gaussian process–based learning algorithm for generalized Hamiltonian systems. The method incorporates three soft constraints—on energy conservation, volume preservation, and Lyapunov stability—to better capture the physical structure and stability properties of the system. Experiments on multiple dynamical systems show that the proposed approach achieves superior accuracy and robustness compared to existing methods.

**Strengths:**

- The authors propose a learning algorithm based on Gaussian processes for the generalized Hamilton systems and introduce three soft constraints to make learning more effective. This aspect appears to be novel.
- The authors verify the effectiveness of the proposed method in multiple dynamical systems, demonstrating its robustness particularly in long-term prediction.

**Weaknesses:**

- The proposed learning process merely adds three regularization terms representing physical constraints to the conventional SSGP method, and thus appears to offer no significant technical contribution.
- The experimental results are encouraging, but a more thorough analysis is required, including consideration of uncertainties.

**Questions:**

- The motivation for introducing three soft constraints needs to be clarified. SSGP is learned to follow Hamilton's equations. Since SSGP  represent Hamilton vector fields, they guarantee at least energy and volume conservation. Therefore, soft constraints on these properties (at least energy and volume) may not be necessary when modeling vector fields. On the other hand, considering learning from trajectory data, cumulative errors due to numerical integration are introduced. Soft constraints may be effective in absorbing this error. Please add the above discussion to clarify the motivation for the proposal.
- After learning, what values did the terms in equation (11) and the hyperparameters take?
- One of the advantages of Gaussian processes is their ability to handle uncertainty, so it would be better to demonstrate their effectiveness in this regard through experiments.
- Please define J and D.

---

### Official Review · Reviewer_nvZF · 2025-10-31

**Soundness:** 3
**Presentation:** 3
**Contribution:** 3
**Rating:** 6
**Confidence:** 2

**Summary:**

This paper proposes a unified and probabilistic framework for learning generalized Hamiltonian dynamics (conservative, dissipative, and port-Hamiltonian) from noisy and sparse trajectory data. The core of the method is to model the Hamiltonian function as a probabilistic surrogate using a sparse Gaussian Process approximated with Random Fourier Features (RFF). To ensure physical plausibility and improve long-term stability, the authors introduce a multi-term loss function that combines the standard evidence lower bound (ELBO) for data-fitting with soft regularization terms enforcing energy conservation, phase-space volume conservation, and Lyapunov-style stability. A key contribution is the use of a Gradient Descent-Ascent (GDA) algorithm to automatically balance the weights of these loss terms, treating it as a min-max optimization problem. Experiments on several benchmark Hamiltonian systems demonstrate that the proposed method achieves superior performance in both short-term accuracy and, most notably, long-horizon forecasting compared to state-of-the-art baselines.

**Strengths:**

- Principled and Unified Framework: The paper presents a unified approach to a complex problem. By parameterizing the three distinct classes of Hamiltonian dynamics within a single RFF-based GP framework, the authors provide a systematic way to tackle a broad range of physical systems. The probabilistic nature of the model is well-justified and naturally handles the challenges of noisy and sparse observations.

- Novel Automated Loss Balancing: The use of GDA to automatically learn the Lagrangian multipliers (λ) is a significant contribution. Balancing multiple, often competing, loss terms is a notoriously difficult hyperparameter tuning problem. The proposed min-max optimization framework offers a principled and automated solution, which enhances the method's practicality and robustness. This is a valuable technique that could be adopted in other multi-task or physics-informed learning settings.

- Comprehensive Experimental Validation: The authors conduct a thorough empirical evaluation across all three classes of Hamiltonian systems. The comparison with strong baselines (HNN variants, SSGP) is fair and clearly highlights the benefits of the proposed method. The ablation studies on noise levels and individual loss components further strengthen the paper's claims and provide valuable insights into the model's behavior.

**Weaknesses:**

- Scalability Concerns: While RFFs improve the scalability of GPs, the experiments are conducted on relatively low-dimensional systems (1D or 2D position spaces). It is unclear how the computational cost and performance of the method, particularly the GDA optimization, would scale to higher-dimensional phase spaces (e.g., many-body systems) or datasets with very long trajectories. A discussion on the computational complexity with respect to the phase space dimension d and the number of RFF features M would be beneficial.

- Stability and Nuances of GDA: The GDA for min-max optimization can be notoriously tricky to train and may not always converge to a desirable equilibrium. The paper mentions this but could benefit from a more in-depth discussion. For instance, in Table 1, the "Ours (Equal)" variant sometimes slightly outperforms the "Ours (GDA)" variant. This raises a question about the stability and reliability of the GDA optimization. Is it sensitive to learning rates or initialization? When and why might a simpler weighting scheme be sufficient or even better?

- Assumptions on D and F(t): The framework makes simplifying assumptions about the structure of the dissipation matrix D (diagonal, only affecting p) and the external force F(t) (also only affecting p). While reasonable for the chosen benchmarks, this limits the generality of the approach. A brief discussion on how the framework could be extended to handle more complex, unknown, or state-dependent dissipation and forcing structures would strengthen the paper.

**Questions:**

- Regarding the GDA balancing: Could you comment on the training dynamics and stability of the GDA approach? As noted in the weaknesses, the equally-weighted version sometimes outperforms the GDA-balanced one in Table 1. Does this suggest that the GDA optimization is sometimes getting stuck in a suboptimal local minimum, or that for some tasks, a simpler balance is more effective?

- Regarding the choice of constraints: The paper applies the same set of regularizers (Energy, Vol, Lyap) across all system types, though their physical meaning changes (e.g., energy is conserved in one case and dissipated in others). The implementation detail "the conservative laws can be enforced by integrating the conservative part only" is key. Could you elaborate on this in the main text? How exactly is L_Energy (Eq. 8) adapted for dissipative and port-Hamiltonian systems, where the total energy is not expected to be conserved? Is it applied only to the J∇H component of the flow?

- Regarding the Lyapunov loss (L_Lyap, Eq. 10): The paper states that for Hamiltonian settings, you can take V=H and α=0. However, the loss term ReLU(d/dt H(x(t))) penalizes any increase in energy. For a port-Hamiltonian system with external energy input, the energy H is expected to increase. How is this apparent contradiction handled? Does the GDA mechanism learn to down-weight this loss term (λ_1 -> 0) in such cases?

---

### Official Review · Reviewer_PVTK · 2025-10-31

**Soundness:** 2
**Presentation:** 2
**Contribution:** 2
**Rating:** 2
**Confidence:** 3

**Summary:**

In this work, the authors propose a probabilisitic framework for learning generalized Hamiltonian dynamics from noisy trajectory data. The method is based on symplectic Gaussian processes with random Fourier features and trained with variational Bayesian inference, where the training loss is augmented by regularizations for enforcing soft physical constraints such as energy conservation, volume conservation and Lyapunov stability and can be solved numerically via gradient descent-ascent (GDA). Experiments demonstrate that the method outperforms prior approaches on a few basic Hamiltonian systems including conservative, dissipative and externally-forced ones.

**Strengths:**

The learning of dissipative and externally forced Hamiltonian systems within a probabilistic framework is novel to my knowledge.

**Weaknesses:**

1. The main novelty of the proposed method compared to prior works such as Tanaka et al. (2022) and Ross and Heinonen (2023) seems to lie in the several penalty terms for softly enforcing the respective physical constraints, together with a min-max formulation of the optimization problem for balancing the different loss terms. It is a bit limited for a publication in ICLR, in my opinion.

2. Does the energy conservation constraint actually make sense when we consider dissipative and externally-forced Hamiltonian systems?

3. My understanding of port-Hamiltonian systems is that they usually refer to interconnected networks of subsystems with force exchanges, which are different from what the authors consider in this paper. It may be clearer to name them as "forced" or "externally-forced" Hamiltonian systems instead.

4. The presentation of the paper, including the quality of the figures, can be improved. The analysis of the experiment results is also rather limited compared to prior works such as Tanaka et al. (2022).

**Questions:**

See item 2 in the weakness section above.

---

### Official Review · Reviewer_iCCD · 2025-11-06

**Soundness:** 3
**Presentation:** 2
**Contribution:** 2
**Rating:** 2
**Confidence:** 3

**Summary:**

The paper proposes a Gaussian process framework for learning different kinds of non-conservative Hamiltonian systems. The main idea is to use a relaxed Hamiltonian framework, and control the model by regularisation towards conservation and stability.

**Strengths:**

The paper is sufficiently original by consider generalised Hamiltonian systems. However, the GP methodology is quite basic, and the overall Hamiltonian GP approach has already been established in earlier works.

The clarity of the paper is overall good, and the paper is easy to follow.

The results show superior performance over baselines, which is a good achievement.

**Weaknesses:**

The math presentation could be improved. The ELBO and the joint distribution is quite oddly presented: the joint distribution has no connection between x_ij and x_0, or any connection between W and x. There is also no theta. I don't think this notation is sufficiently rigorous.

The paper oddly shows no system fits in the main paper, which makes it quite difficult to get a good intuition on what is happening, how much data is used, or what do the predictions look like. In appendix there are visualisations, which seems to show quite different picture from the tables. Figs 4-6 all show that the there is basically no difference between SSGP and the proposed method (!), and the GP methods make really strange and really strong error patterns. I suspect that there is something wrong in the implementation of this method, or there are some serious misidentification issues in these models.

Finally, it's difficult to see in what ways the proposed method is significant. There is no "real-world" usecase, and all the experiments are small-dimensional simple systems that we probably could model better by conventional means.

**Questions:**

See above

---

### Note · Authors · 2025-12-03

I have read and agree with the venue's withdrawal policy on behalf of myself and my co-authors.